# Comparative Analysis of Properties of PVA Composites with Various Nanofillers: Pristine Clay, Organoclay, and Functionalized Graphene

**DOI:** 10.3390/nano9030323

**Published:** 2019-03-01

**Authors:** Jin-Hae Chang

**Affiliations:** Department of Polymer Science and Engineering, Kumoh National Institute of Technology, Gumi 39177, Korea; changjinhae@hanmail.net

**Keywords:** poly(vinyl alcohol), nanocomposite, nanofiller, film

## Abstract

Poly(vinyl alcohol) (PVA) nanocomposites containing three different nanofillers are prepared and compared in terms of their thermal properties, morphologies, and oxygen permeabilities. Specifically, pristine saponite (SPT) clay, hydrophilic organically modified bentonite (OMB), and hexadecylamine-functionalized graphene sheets (HDA-GSs) are utilized as nanofillers to fabricate PVA nanocomposite films. The hybrid films are fabricated from blended solutions of PVA and the three different nanofillers. The content of each filler with respect to PVA is varied from 0 to 10 wt%, and the changes in the properties of the PVA matrices as a function of the filler content are discussed. With respect to the hybrid containing 5 wt% of SPT, OMB, and HDA-GS, each layer in the polymer matrix consists of well-dispersed individual nanofiller layers. However, the fillers are mainly aggregated in the polymer matrix in a manner similar to the case for the hybrid material containing 10 wt% of fillers. In the thermal properties, SPT and OMB are most effective when the filler corresponds to 5 wt% and 7 wt% for HDA-GS, respectively, and the gas barrier is most effective with respect to 5 wt% content in all fillers. Among the three types of nanofillers that are investigated, OMB exhibits optimal results in terms of thermal stability and the gas barrier effect.

## 1. Introduction

Poly(vinyl alcohol) (PVA) is a water-soluble synthetic polymer with high hydrophilicity, biocompatibility, and non-toxicity [1,2]. The high capacity of PVA to simultaneously form both intra- and inter-chain hydrogen bonds make it a unique polymer that can interact with nanofillers such as clay, graphene, and functionalized-graphene [3,4]. For example, incorporation of graphene oxide (GO) into PVA facilitates good dispersion and interfacial interaction due to the presence of OH-bonds at the end due to the interaction of OH bonding. The content of clay can readily affect the thermo-mechanical properties and gas permeability [5,6]. 

Clay and graphene exhibit high aspect and high load transfer results in the agglomeration of layers in polymer matrix, and this makes dispersion difficult. In order to avoid this practical challenge, it can be exfoliated or organically modified to achieve the desired properties by dispersing nanofillers in compatible solvent via sonification [7].

In the wet state or especially after mild drying, clay layers are distributed and embedded in the PVA gel to yield a true nanoscale hybrid material. However, drying in vacuo can cause the re-aggregation of the clay layers. The steric constraints created by the PVA matrix impede the re-aggregation of the clay layers, and, thus, a few clay layers remain in the dispersed state. Ideally, useful nanocomposites are fabricated to create amorphous domains with uniformly distributed mineral layers. However, the preparation of PVA/clay nanocomposite materials from a solution is challenging due to the re-aggregation of the layers [8].

Clays with sandwich-type structures that typically consist of an octahedral Al sheet and multiple tetrahedral Si sheets are referred to as phyllosilicates [9,10]. There are several types of phyllosilicates including kaolinite, montmorillonite, hectorite, saponite, bentonite, and synthetic mica. In the present study, we selected saponite (SPT) [11,12] and hydrophilic organically modified bentonite (OMB) [13,14] as clays for the synthesis of clay/PVA polymer nanocomposites. Specifically, SPT and bentonite consist of stacked silicate 1-nm-thick sheets with lengths of approximately 165 nm and 68 nm, respectively. Currently, SPT and bentonite are widely employed as reinforcing fillers in polymeric matrices due to their excellent mechanical, electrical, and thermal properties and their low cost [15,16,17].

Generally, with the exception of SPT, several types of pristine clays are not compatible with most polymers, and, thus, require organic treatments to render them as organophilic. A common method for this type of organic treatment is based on the ion exchange of the cations within the clay with organic ammonium cations [18,19,20]. Thus, we selected hydrophilic organo-clay OMB as nano-filler for the synthesis of hybrid polymer films.

Graphene tends to aggregate or restack due to its strong stacking tendency and high cohesive energy. Additionally, it is insoluble in a variety of organic solvents due to its significantly hydrophobic nature and high specific surface area of graphene. Therefore, a key challenge in the preparation and processing of graphene-based composites corresponds to the prevention of aggregation. The functionalization of the graphene surface can introduce reactive moieties that disrupt the bundle structure and can potentially obtain individual sheets [21,22,23]. This type of functionalization involves the attachment of functional moieties to the open ends and walls of graphene to improve the solubility and dispersibility of graphene sheets (GSs) [24]. Hence, an optimal method to achieve a homogeneous graphene dispersion throughout a polymer matrix is the use of functionalized graphene sheets (FGSs) [25] that exhibit improved dispersion in solvents and polymers. Furthermore, covalent functionalization can provide the means to engineer the GS/polymer interface and, thereby, optimize the properties of the composite material.

Typically, traditional composite structures typically contain a significant content (≈ 40 wt%) of a filler bound within a polymer matrix. However, significant changes in the properties of the materials are also possible at low loadings (< 10 wt%) of nano-fillers, such as exfoliated pristine clays, organoclays, and FGSs, in the hybrid materials [26,27]. The improvements in material performance are achieved as a result of the inherent properties of the nanofillers and also by optimizing the dispersion, interface chemistry, and nanoscale morphology. This is completed to utilize the advantages of the tremendous surface area per unit volume exhibited by nano-fillers (the theoretical limits correspond to 760 m^2^/g for clay [28] and 2,630 m^2^/g for graphene [29]).

In the present study, we prepared hybrid films containing PVA and an appropriate amount of filler (≤ 10 wt%) and examined their properties as a function of the filler content and type. We examine and compare the thermal properties, morphologies, and oxygen permeation capabilities of PVA nanocomposites containing three different nanofillers, such as pristine clay SPT, hydrophilic organoclay OMB, and hexadecylamine-functionalized GSs (HDA-GSs). The thermal and oxygen barrier properties of the hybrids are also examined as a function of the nanofiller type and content in the PVA polymer matrix. Lastly, we investigate the effects of filler loadings on the morphologies of the PVA hybrid films. 

## 2. Materials and Methods

### 2.1. Materials

The source clays, SPT, and OMB were obtained from Kunimine Ind. Co. (Tokyo, Japan) and Nanomer Co. (Seoul, Korea), respectively. The clays were passed through a 325-mesh sieve to remove impurities to yield an SPT clay with a cationic exchange capacity of 100 meq/100 g and bentonite with a cationic exchange capacity of 145 meq/100 g. Additionally, PVA with >99% saponification (*M*_w_ = approximately 89,000–98,000), graphite, and HDA were purchased from Aldrich Chem. Co. These materials were used in an as-received condition. Commercially available solvents were purified via distillation, and common reagents were used without further purification.

### 2.2. Synthesis of HDA-GS

Graphene oxide (GO) was synthesized from natural graphite using a multi-step route known as the Hummers method [30]. Furthermore, HDA-GS was synthesized from hexadecylamine (HDA) and GO based on the following procedure: GO (1 g) was dissolved in 1.5 L of distilled water. HDA (2.00 g; 8.28 × 10^−3^ mol) was added to 25 mL of ethanol, and the mixture was stirred at 25 °C under a steady stream of N_2_, and subsequently added to the GO/water mixture. The resulting mixture was heated for 12 h at 25 °C under a steady stream of N_2_, cooled to 25 °C, washed twice with a mixture of distilled water and ethanol (1:1, v/v), and dried under vacuum at 70 °C for 24 h to obtain HDA-GS. The synthetic route for HDA-GS is shown in Scheme 1.

### 2.3. Preparation of PVA Hybrid Films

The synthetic procedures used to produce the polymer hybrids were identical for all filler contents used in this experiment. Therefore, the preparation of 5-wt% SPT/PVA is detailed in this paper as a representative example. Specifically, SPT (0.1 g) was added to distilled water (20 mL) in a 100-mL beaker, and the mixture was stirred at 80 °C for 1 h. The resulting mixture was subjected to ultrasonication three times for 5 min to obtain a homogeneously dispersed clay solution. In a separate beaker, PVA (1.9 g) and distilled water (140 mL) were mixed at 80 °C for 3 h and subsequently added dropwise to the SPT/water system with vigorous stirring for 3 h to obtain a homogeneously dispersed system. The solution was cast on poly(ethylene terephthalate) (PET) films and evaporated in a vacuum oven at 35 °C for two days. After the removal of the solvent, the hybrid film was dried for a second time in a vacuum oven at 80 °C for a day.

### 2.4. Characterization

Fourier-transform infrared (FT-IR) spectra were obtained via an FT-IR 460 (JASCO, Tokyo, Japan) instrument in the range of 4000 to 600 cm^−1^ with KBr pellets. Wide-angle X-ray diffraction (XRD) measurements were performed at room temperature via a Rigaku (D/Max-IIIB, Tokyo, Japan) X-ray diffractometer with Ni-filtered CoK_α_ radiation. The scan rate corresponded to 2°/min over the 2*θ* range of 2° to 10°. Differential scanning calorimetry (DSC 200F3, Berlin, Germany) was performed on a NETZSCH instrument, and a thermogravimetric analyzer (AutoTGA 1000, New Castle, USA)) was employed as the thermogravimetric analysis instrument with a heating rate of 20 °C/min under the flow of N_2_.

Atomic force microscopy (AFM, Multimode, NanoScope III, Digital instruments Inc. NY, USA) images were obtained on an AutoProbe CP/MT scanning probe microscope. The GO samples were dispersed in water and HDA-GS samples in toluene. The suspensions were ultrasonicated for 3 h and subsequently spin-coated at 5000 rpm on silicon wafers. 

The morphologies of the fractured surfaces of the extrusion samples were investigated via a Hitachi S-2400 scanning electron microscope (SEM). In order to enhance the conductivity, the fractured surfaces were sputter-coated with gold via an SPI sputter coater. Transmission electron microscopy (TEM) samples were prepared by placing the PVA hybrid films on epoxy capsules and curing them at 70 °C for 24 h under vacuum. The cured epoxy capsules containing the PVA hybrids were microtomed into 90-nm thick slices and positioned on a 200-mesh copper net, and a layer of carbon (approximately 3-nm thick) was deposited on each slice. The TEM images of the ultrathin sections of the polymer hybrid samples were obtained via an EM 912 OMEGA TEM instrument with an acceleration voltage corresponding to 120 kV.

The O_2_ transmission rates (O_2_TRs) of the films were measured based on ASTM E96 with a Mocon DL 100 instrument. The O_2_TRs were obtained at 23 °C, 0% relative humidity, and 1 atm pressure. 

## 3. Results and Discussion

### 3.1. FT-IR Spectroscopy

Figure 1 shows the FT-IR spectra of PVA, graphite, GO, HDA-GS, and OMB. The spectrum of pure graphite does not exhibit any peaks while that of PVA and GO exhibit significant and broad absorption peaks characteristic for the OH and COOH functional groups [31]. Details of each substance are specified below.

The characteristic absorption peaks of the PVA are observed at 3330 cm^−1^ (O–H stretching), 2940 cm^−1^ and 2910 cm^−1^ (asymmetric stretching CH_2_), 1730 cm^−1^ (due to water absorption), and 1256 and 1090 cm^−1^ (C-H bending and C-O stretching). In the case of the GO, the characteristic absorption peaks of the O-H are observed at 3220 cm^−1^ (stretching) and asymmetric epoxy appears in 3063 cm^−1^, but does not appear to overlap with OH, 1732 and 1616 cm^−1^ (C=O stretching), and 1040 cm^−1^ (C–O stretching). The epoxide ring and C=C bond are much weaker compared to the others, whereas the O–H stretching is more intense. Figure 1 also shows the spectrum of HDA-GS: 3160 cm^−1^ (O–H stretching), 2921 and 2852 cm^−1^ (aliphatic C–H stretching), 1375 cm^−1^ (aromatic C–N–C symmetric stretching), and 790 cm^−1^ (N–H out-of-plane stretching). The OMB also exhibits a peak at 1100 cm^−1^ (C-O stretching).

### 3.2. Morphology of HDA-GS 

The SEM images of unmodified natural graphite, GO, and HDA-GS are shown in Figure 2. The natural graphite exhibits a lamellar structure similar to that of graphene sheets with stacks denser than those observed for the other materials (Figure 2a). Figure 2b shows the translucent GO sheets that are wrinkled and folded in a manner resembling thin paper. Additionally, HDA-GS is prepared from chemically modified GO and exhibit an entangled morphology and a random distribution (Figure 2c).

Generally, a significant volume expansion and high porosity are observed in the FGSs, which results in low FGS bulk densities that can cause feed problems during melt compounding. Normally, master batches are required to solve the problem. However, in the study, solution blending of FGSs with polymers is successfully employed.

The AFM imaging provides more reliable information on the sheet dimensions and can also be used to probe the surface topology, defects, and bending properties. Additionally, stepped-height scans can also allow us to determine the lateral sizes and thicknesses of the particles lying on the substrates. Thus, AFM is used to indicate that the carbon sheets obtained in the present study are comprised of only a single atomic layer. Figure 3 shows the AFM image of HDA-treated GO sheets (HDA-GSs) on mica and a profile plot that reveals the average sheet thickness corresponding to 1.76 nm (Figure 3) when the thickness of the bare graphene sheet is approximately 1 nm [32], and the thickness of the layer of substituted HDA organic groups is approximately 0.76 nm. However, the layer of the substituted HDA organic groups exhibits a thickness of approximately 0.38 nm when the organic HDA groups on both sides of the graphene sheet in the synthesized FGS are significantly tilted. The change in thickness is associated with different orientations that are adopted by the long alkyl chains in the chemically modified HDA-GSs.

### 3.3. Dispersion

The XRD traces of the pure nanofillers and their PVA hybrid films are shown in Figure 4. The *d*_001_ reflection of pristine SPT is present at 2*θ* = 6.62°, and this corresponds to an interlayer spacing *d* of 13.54 Å, as shown in Figure 4a. Specifically, Figure 4a also shows the XRD curves of the SPT/PVA hybrid films with clay contents in the range of 0–10 wt%. With respect to the PVA hybrids with a clay content ≤10 wt%, clay peaks did not appear in the XRD traces, which indicates that the clay particles are homogeneously dispersed in the hybrid polymer matrices. Figure 4b also shows the XRD curves of OMB in the region 2*θ* = 2°–10°. The *d*_001_ reflection of OMB is observed at 2*θ* = 6.64°, and this corresponds to an interlayer distance (*d*) of 13.30 Å. In the case of PVA hybrids containing 3 wt% of OMB, only a very slight peak appeared at 2*θ* = 4.78°, and this corresponds to an interlayer spacing of 18.46 Å (Figure 4b). The result indicates that a small amount of the clay is not aggregated in the PVA matrix. However, a significant increase in aggregation is observed for samples with OMB loadings reaching 10 wt%, as shown by the intensities of the XRD peaks. This suggests that perfect exfoliation of the layered structure of the clay did not occur. Further evidence of clay dispersion into PVA on a nanometer scale is obtained via TEM.

The XRD diffractograms of pure HDA-GS and HDA-GS/PVA hybrid films are shown in Figure 4c. The *d*_001_ reflection for HDA-GS is observed at 2*θ* = 2.73° and corresponds to an interlayer spacing (*d*) of 32.32 Å. With respect to PVA hybrid films containing up to 7 wt% of HDA-GS, the peak observed at 2*θ* = 2.73° for GS is observed to disappear from the diffraction patterns. The result indicates that the graphene layers are exfoliated and homogeneously dispersed throughout the PVA matrix and provide supporting evidence for the nanocomposite character of the HDA-GS/PVA hybrids. However, the intensities of the XRD peaks at 2*θ* = 2.64 (*d* = 33.42 Å) and 2*θ* = 5.35 (*d* = 16.50 Å) increase suddenly when the HDA-GS loading increases from 7 wt% to 10 wt%, which suggests that the dispersion is more effective at lower loadings as opposed to higher loadings of graphene.

Given the periodic arrangement of the graphite layers in the virgin GS and in intercalated hybrids, the XRD offers a convenient method to determine the interlayer spacing. However, although the XRD enables precise routine measurements of the GS layer spacings, it neither allows for the determination of the spatial distributions of GSs or the detection of any inhomogeneous sections of the hybrids. Initially, a few layered GSs do not exhibit well-defined basal reflections, and, thus, it is difficult to systematically follow peak broadening and reductions in intensity. Therefore, all conclusions on the mechanisms of hybrid formation and microstructure, based solely on XRD results, are only tentative. Therefore, further evidence of the GS dispersion in the PVA films on a nanometer scale is obtained via SEM and TEM as described in the next section.

### 3.4. Morphologies of PVA Hybrids

In addition to using XRD to measure the *d* spacings of the nanocomposites, SEM and TEM are used to evaluate the degree of intercalation and amount of aggregation in the nanofiller clusters. The morphologies of the aggregated fillers are characterized via SEM. Large filler aggregates can be easily imaged by SEM due to the difference between the scattering densities of the filler and the PVA matrix [33].

The morphologies of the hybrid films containing up to 10 wt% of SPT in the PVA matrix are examined by observing their fracture surfaces by SEM (Figure 5a). The PVA hybrid films containing 3 wt% and 5 wt% of SPT display uniform and dispersed phases. Conversely, the films containing 7 wt% and 10 wt% of SPT exhibit large particles and some deformed regions that can result from the coarseness of the fractured surface. A similar type of behavior was observed for PVA/OMB hybrid films. For example, the fractured surface of the 5-wt% OMB hybrid film (Figure 5b) exhibit uniform and dispersed phases. However, increased agglomeration is observed in the PVA matrix with clay content exceeding 7 wt% in the OMB system.

The fractured surfaces display increased levels of deformation for samples with higher clay contents. The trend is most likely linked to increases in the agglomeration of clay particles and indicate the lack of interfacial interactions between the clay and matrix polymers. Thus, several defects and significant agglomeration occur in interphase areas in high clay content nanocomposite PVA films. Figure 6 shows a comparative analysis of the SEM micrographs obtained for the PVA hybrids with different contents of HDA-GSs that exhibit platelet-orientation distribution morphology.

Graphene dispersions are readily observed in the SEM images due to the differences in the scattering densities of graphene and the matrix polymer. The SEM images of the fractured surfaces of PVA hybrid films containing 0–10 wt% of HDA-GS are compared in Figure 6. The hybrid film with 5 wt% of HDA-GS display a morphology consisting of graphene domains that are well dispersed throughout the continuous PVA phase. However, deformed surfaces and voids are observed in the case of the 7-wt% FGS hybrid film. Conversely, the micrographs of the 10-wt% HDA-GS/PVA hybrid films exhibit increased levels of voids and deformed regions when compared to the 3–7 wt% HDA-GS/PVA hybrid films due to the coarseness of the fractured surface. Overall, the comparison reveals that increases in the graphene content in the hybrid films increase the level of deformation of their fractured surfaces. The finding potentially results from the agglomeration of graphene particles. A comparison of the micrographs indicate that the fractured surfaces of the hybrid films with higher graphene content are more deformed than those of hybrid films with lower graphene content, and this is possibly due to the agglomeration of graphene particles [34,35]. It should be noted that Figure 6 also shows that most of the graphene remains in the form of straight and rigid platelets in the composite, which indicates that the graphene sheets are extremely stiff. 

We extend the morphological analysis via TEM to evaluate the degree of intercalation and degree of aggregation of the nanofiller clusters. Additional direct evidence for the formation of a true nanocomposite is provided by the TEM analysis of ultra-microtomed sections. Figure 7 shows the micrographs of the PVA hybrid films with identical contents of the three different nanofillers. The dark lines in the photographs denote the intersections of the clays and GSs (1-nm thick) while the length between the dark lines denotes the interlayer distance. Figure 7a shows the morphologies of the PVA hybrids with 5 wt% and 10 wt% of SPT. With respect to the hybrid containing 5 wt% of SPT, each layer in the polymer matrix consists of well-dispersed individual clay layers (dark lines), and a few of the clays aggregate to a thickness of approximately 10 nm. In a manner similar to the case for the hybrid material containing 10 wt% of SPT, these clays are mainly aggregated in the polymer matrix. However, the average particle size is observed to be below 20 nm, as calculated from the TEM images. The presence of agglomerated particles in the SEM micrographs of hybrid materials with higher SPT contents (see Figure 5a) is attributed to the formation of aggregated layers.

Typical TEM images of PVA hybrid films containing 5 and 10 wt% OMB are shown in Figure 7b. Evidently, the clays are well dispersed within the polymer matrix irrespective of the clay content. In contrast to the clay layers in the hybrids containing SPT, the clay layers in OMB hybrids are exfoliated within the matrix polymer. 

The TEM micrographs of the 5 and 10 wt% HDA-GS hybrid films are shown in Figure 7c. The TEM micrographs indicate that the GS in the 5 wt% HDA-GS hybrid is dispersed in the polymer matrix, which indicates the formation of a nanocomposite. The findings suggest that GS breaks down into nanoscale building blocks during the intercalative polymerization process and is homogeneously dispersed in the polymer matrix to yield a polymer/GS nanocomposite. As in the case of the 10 wt% HDA-GS (see Figure 7c), the GSs are mostly agglomerated in the polymer matrix. In contrast to the hybrids containing 5 wt% of HDA-GS, the graphene layers of the 10 wt% hybrid exhibited agglomeration of the dispersed graphene phase and are not intercalated into the matrix polymer. The agglomeration of the dispersed graphene phase visibly increases with increases in graphene content, and the outcome is consistent with the XRD and SEM data shown in Figure 4c and Figure 6.

The XRD, SEM, and TEM results indicate that the fillers are well dispersed throughout the PVA matrix at low filler contents while aggregated structures are present at higher filler contents. Additionally, the dispersions of SPT and OMB exceed that of HDA-GS in the PVA matrix (see Figure 4, Figure 5, Figure 6 and Figure 7). The unusual thermal and gas barrier properties of these hybrid films are discussed in the following sections with respect to the dispersion of the nanofillers.

### 3.5. Thermal Properties

A comparison of the DSC results for pure PVA and PVA hybrids with approximately 3–10 wt% of clays (SPT and OMB) and graphene (HDA-GS) are listed in Table 1. The glass transition temperature (*T_g_*) of pure PVA corresponds to 69 °C. The *T_g_* values of PVA hybrids containing various clay contents are virtually unchanged in the DSC results when compared to that of pure PVA irrespective of the filler loading, i.e., approximately 68–71 °C for the three different fillers. Generally, *T_g_* increases when the filler content increases up to a critical concentration. The increase in *T_g_* is potentially due to the confinement of the intercalated polymer chains within the filler galleries that prevents the segmental motion of the polymer chains [36,37]. However, the *T_g_* values of the three types of PVA hybrid films remains constant irrespective of the filler content, which suggests that the variation in the filler content does not affect the confinement of the PVA chains.

The endothermic peak of pure PVA appears at 165 °C and corresponds to its melting transition temperature (*T_m_*) (see Table 1). The *T_m_* values of the hybrid films are observed to increase from 165 to 176 °C when the SPT loading is increased from 0 to 5 wt% and, subsequently, decreases to 166 °C at an SPT content of 10 wt%. The increase in the *T_m_* of the hybrid film potentially occurs as a result of the insulation effect of the clays and the interactions between the clay and PVA chains [38]. However, the decrease in the *T_m_* value of the 10-wt% SPT hybrid suggests that its domains are more poorly dispersed in the PVA matrix than those in the 5-wt% SPT hybrid. Hence, increases in clay content led to the aggregation of clay particles, and this reduces the heat-insulation effect of the clay layers in the polymer matrix.

A similar trend is observed for both the OMB and HDA-GS hybrids. Specifically, the *T_m_* of the PVA hybrids increases from 165 to 184 °C and from 165 to 173 °C when the filler loadings increase to 5 wt% for OMB and 7 wt% for HDA-GS. As observed for the SPT hybrid material, the maximum transition peaks of the PVA hybrids increase with the addition of the nanofiller only up to a certain content level and, subsequently, decreases when the content is increased above this point. For example, when the filler content of PVA reaches 10 wt%, the *T_m_* decreases to 178 °C and 156 °C for materials employing OMB and HDA-GS nanofillers, respectively. The DSC thermograms of the PVA hybrids with various HDA-GS contents are shown in Figure 8. When the nanofiller is included, the peak is stronger than that of the pure PVA, and the more the amount of the filler increased up to 7 wt%, the greater the intensity of the peak becomes. Hence, the HDA-GS appears to act as a nucleating agent [39]. However, when the amount of HDA-GS reaches 10 wt%, the degree of dispersion decreases in conjunction with decreases in the peak intensity.

In a manner similar to the results for *T_m_*, the initial thermal degradation temperatures (*T_D_^i^*) of the PVA hybrid films also increases linearly from 227 °C to 249 °C while increasing the SPT loading from 0 to 5 wt%, as shown in Table 1. With respect to the hybrids containing OMB, the *T_D_^i^* value varies from 227 to 252 °C when the content of the OMB organoclay increases from 0 to 5 wt% in the PVA hybrids. The highest increase of 25 °C in the *T_D_^i^* relative to that of pure PVA is observed for the 5 wt% OMB hybrid (252 °C). The clay enhances the *T_D_^i^* by acting as an insulator and a mass-transport barrier to the volatile products generated during decomposition [40,41]. The increase in thermal stability is attributed to the high thermal stability of the clay and interactions between the clay particles and polymer matrix. In contrast to the behavior observed for clay contents ranging from 0 to 5 wt%, the *T_D_^i^* values of the hybrids decreases when the clay content increases from 5 to 10 wt%. For example, the *T_D_^i^* values of PVA hybrid film containing 10 wt% of clay loadings were 11 °C (238 °C) and 6 °C (246 °C) lower than those of PVA hybrids containing 5 wt% of SPT and OMB, respectively. The decrease in *T_D_^i^* appeared to correspond to the result of clay aggregation that occurs when the clay content in the polymer matrix exceeds a critical value. Similar results are observed in samples containing HDA-GS, as shown in Table 1. The *T_D_^i^* values of the PVA hybrid films also increases linearly from 227 to 245 °C when the HDA-GS loading is increased from 0 to 7 wt%. At 10 wt% content of HDA-GS in the PVA, the *T_D_^i^* decreases again to 238 °C. 

An analysis of the weight residue at 600 °C (wt_R_^600^) indicates that the weight increases with growth in the filler loading from 0 to 10 wt% and corresponds to a range from 3% to 15% for SPT, from 3% to 14% for OMB, and from 3% to 15% for HDA-GS (Table 1). The enhancement in char formation with increases in filler content is ascribed to the high heat resistance of the clays and graphene.

### 3.6. Gas Permeation

The mobility of the polymer chain segments in the polymer nanocomposite clearly differs from that of the pure polymer due to the confined environment, and this also affects the gas permeability. The following two main factors are responsible for permeability reduction [42,43,44]: (i) polymer chain-segment immobility and (ii) detour ratio, which is defined as the ratio of the film thickness in the nominal diffusion flow direction to the average length of the tortuous diffusion distance between nanolayers. 

High aspect ratio nanolayers also lead to properties that are not possible for larger-scaled composites. The impermeable nano-sized layers mandate a tortuous pathway for a permeant to transverse the nanocomposite. Enhanced barrier characteristics, chemical resistance, reduced solvent uptake, and flame retardance of polymer hybrids benefit from the hindered diffusion pathways through the nanocomposite [43,44].

In order to further characterize the barrier properties of the PVA hybrids fabricated by the intercalation of polymer chains in the galleries of clays and graphene, the permeability of the resulting PVA hybrid films to O_2_ is evaluated for various filler loadings in the range of 0–10 wt%. The results are summarized in Table 2. The thicknesses of all the films subjected to the gas permeation measurements are in the range of 20 to 24 μm. We discuss our results in terms of relative permeability, *P_c_/P_p_*, where *P_p_* denotes the permeability of the pure polymer and *P_c_* denotes the permeability of the composite. The results confirm that the mass-transfer process for O_2_ as the penetrant is highly dependent on the level of filler loading. For example, the addition of only 7 wt% of SPT results in a 95% reduction in the permeability rate of O_2_ (0.24 cc/m^2^/day) relative to that of the pure PVA film (5.13 cc/m^2^/day). With respect to the PVA hybrid films containing 3 to 7 wt% of OMB, the relative O_2_ permeability rate is close to zero (see Table 2). The outcome is attributed to the increase in the length of the tortuous paths followed by the gas molecules and interactions between the O_2_ and clay molecules. Furthermore, films containing higher amounts of clay appear as significantly more rigid, and this decreases their gas permeability. In a manner similar to the results observed for the clay hybrids, the addition of 5 wt% of HDA-GS results in an 81% reduction in the permeability rate of O_2_ (0.98 cc/m^2^/day) relative to that of the pure PVA film (5.13 cc/m^2^/day). However, when the clay content increases from 7 to 10 wt%, the O_2_ permeability rate slightly increases from 0.24 to 0.88 cc/m^2^/day for SPT and more significantly from <10^−2^ to 3.18 cc/m^2^/day for OMB. The increase in the HDA-GS loading from 5 to 10 wt% led to a similar increase in permeability from 0.98 to 4.27 cc/m^2^/day. The increases in permeability values are primarily due to the aggregation of the filler particles in materials employing loadings that exceed the critical filler content levels. The present results are further corroborated via the electron micrographs shown in Figure 5, Figure 6 and Figure 7. 

Collectively, the results of the gas permeation analysis reveal that the strongest gas barrier effect of OMB is observed with respect to the O_2_TR among the three types of nanofillers that are studied. The enhanced gas barrier capacity of OMB stems from its hydrophilic character, and this allows the formation of hydrogen bonds between OMB molecules and the PVA polymer matrix as well as the reinforcement of chain packing, which significantly reduces the gas permeability of the material [45].

## 4. Conclusions

In the present study, we investigated the dispersibilities of three nanofillers including pristine clay SPT, hydrophilic organoclay OMB, and functionalized graphene sheets HDA-GSs, in PVA to improve the properties of the PVA hybrid films. Specifically, PVA hybrid films with varying filler contents ranging from 0 to 10 wt% were synthesized via the solution intercalation method. Their thermal properties, morphologies, and gas permeabilities were compared. The present results confirmed that the properties were dependent on the type and quantity of the nano-filler incorporated in the PVA polymer matrix.

The morphologies of the hybrid materials were examined via TEM, which confirmed that OMB exhibited better dispersion properties than SPT and HDA-GS with respect to the PVA matrix. This observation agreed with the thermal stabilities and gas barrier capabilities of these hybrid materials with the same filler loading levels. Furthermore, the results indicated that the addition of a small amount of nanofiller can sufficiently improve the properties of the PVA. Overall, the addition of OMB was more effective than the addition of SPT and HDA-GS to improve the thermal stability and O_2_TR of the PVA hybrid composite due to the interactions that formed between the OMB and hydrophilic PVA.

In summary, we demonstrated a simple and effective method to fabricate PVA nanocomposites using the solution intercalation method. Improvements in the thermal property and gas barrier of the obtained composites were observed. It is expected that the use of the proposed methods will allow the widespread use of PVA hybrids in various applications such as permeation membranes, polymer electrolyte fuel cell, packaging films, and drug delivery. Application of the technique to the nano-sized fillers in other polymer composite materials can enhance the various advantages of polymeric materials. The technique can be utilized to further improve thermo-mechanical properties by using clay and other types of fillers based on carbon.

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
