# Peer review of "Comparative Analysis of Properties of PVA Composites with Various Nanofillers: Pristine Clay, Organoclay, and Functionalized Graphene"

_nanomaterials, 2019, doi:10.3390/nano9030323_

Round 1
Reviewer 1 Report
This paper reports on three PVA films containing nano-materials.
Each addtional function to PVA are important, which is significance.
No problems could be found throughout the manuscript.
So it should be accepted as it is.
That's all.
Author Response
There is no point from the reviewer-1.
Reviewer 2 Report
Even though the manuscript has serious drawbacks, the aim of this investigation is challenging. Consequently, it should be reconsidered before publishing in Nanomaterials. Based on the comments given below, I recommend a Major Revision of the manuscript.
Several critical points are listed below.
1. The English used in this manuscript needs an improvement. There are too many spelling mistakes and some phrases are not clearly written /explained.
2. In the actual form, the abstract is just an enumeration of results. In this respect, the abstract must be rewritten to briefly summarize the purpose of this study, the main results and major conclusions.
3. In the Introduction, the novelty is weakly emphasized. There are plenty of papers already published on PVA-based composites. In this context, the authors’ motivation for selecting this system needs stronger explanations. Furthermore, the list of references should be updated with some articles recently published.
4. In the Experimental part, the composition of the nanofillers should be included.
5. In Figure 1, I recommended only the important bands should be presented and discussed. Also, discussion of each change in band should be supported by references. For clarity, in Figure 1 should be also added the FTIR spectrum of the nanofillers before incorporation into PVA.
6. What is the application of the new PVA composites with various nanofillers prepared in this study?
7. In their current state, the manuscript conclusions are only an enumeration of the methods used for PVA/nanofillers composites characterization. From my point of view, their conclusion should summarize the obtained results and set them into perspective to the objectives formulated in the introduction. Moreover, it must give an outlook on the importance of their findings. So, the authors should reword the conclusions accordingly.
Additional minor points are listed below.
The resolution of Figures 4 and 6 should be improved.
The description of all abbreviations should be added separately.
Finally, I consider that this manuscript presents interesting and valuable data for the scientific audience, and it would be a pity if unclear explanations or the lack in emphasizing their work novelty will diminish its value.
Author Response
Dear Editor
This is my response to your comments regarding our paper “Comparative Analysis of Properties of PVA Composites with Various Nanofillers: Pristine Clay, Organoclay, and Functionalized Graphene (nanomaterials-431778) in nanomaterials.
Thank you very much for the referee's comments. I have carefully revised the manuscript following the comments of the referee.
Response to Reviewer-2 comments
Point-1: The English used in this manuscript needs an improvement. There are too many spelling mistakes and some phrases are not clearly written /explained.
Response-1: As the referee pointed out, the manuscript is re-edited English grammar through a professional institution and re-readed it from the beginning. English editing is attached in the file.
Point-2: The abstract must be rewritten to briefly summarize the purpose of this study, the main results and major conclusions.
Response-2: As pointed out, I explained a detailed description of the briefly summarize the purpose of this study: “With respect to the hybrid containing 5 wt% - with respect to 5 wt% content in all fillers..” was added in the end of Abstract.
Point-3: Deep discussion in Introduction.
Response-3: As pointed out, the contents and applications of PVA and its hybrids have been added and recent references have been supplemented. The PVA was explained in detail in the beginning of the Introduction section. “Poly(vinyl alcohol) (PVA) is a - in compatible solvent via sonification [7].” was added in the beginning of Introduction section. The latest literature is supplemented by references 1-7.
Point-4: The composition of the nanofillers.
Response-4: The exact chemical composition of clay and GO is not yet known. Therefore, it is not possible to know the exact chemical composition of the organo clay and functionalized graphene using them. It is only guessing with reactivity and physical results.
Point-5: Discussion of each change in band and give the peak assign in FT-IR
Response-5: Figure 1 supplements the peaks for PVA and OMA and provides a detailed description of each peak. “Figure 1 shows the FT-IR spectra of - a peak at 1100 cm−1 (C-O stretching).” was added in Section A. FT-IR spectroscopy.
Point-6: The application of the new PVA composites with various nanofillers
Response-6: In the conclusion section, I explained the application field to use the PVA nanocomposite obtained using the nanofiller. “In summary, we demonstrated - and other types of fillers based on carbon.” was supplemented in the end of Conclusion Section.
Point-7: Reinforcement of Conclusion
Response-7: I supplemented the conclusions already mentioned in the above 6.
Additional minor points are listed below.
Point-8: The resolution of Figures 4 and 6 should be improved.
Response-8: If it's published, it'll replace it with a better picture. I'm working on it, but if it's difficult, I'll do it again.
Point-9: The description of all abbreviations should be added separately.
Response-9: I supplemented the abbreviation part as indicated after the Conclusion section.
I hope this revision is satisfactory for your further process.
Best,
Jin-Hae Chang
Professor

Reviewer 3 Report
The authors show the preparation and characterization of some nanocomposites, but some spectroscopic data can be improved to increase the readability and understanding.
The paper can be improved, and this version is not recommended for publication. Some major revisions are necessary to modify the manuscript:
1)the amount of 10% by wt is very huge for a nanocomposite by using a very polar polymer matrix such as PVOH. The maximum content od nanocomposites should be 5 % by wt, but many previous papers reported that the best solution is 3% by wt in agreement with the reported results. Some experimental data are redundant! A content of 3% by wt of nanocomposites are used often as masterbatches for a further dilution, which usually decrease aggregation domains with the maintenance of the permeability, thermal and morphological properties.
2)Experimental techniques are well presented and discussed. However, DSC data should be evidence that the nucleating effect increasing the crystallization temperature. Unfortunately, these data are not reported and discussed. It very useful for the reader to have a figure with the DSC thermograms!
3) the GPO modification followed by FTIR is not clearly described and reported. The absorption band of epoxy groups is not necessarily formed and present on the GPO. Are there some experimental evidence or is only a reasonable hypothesis? Furthermore, the modification reaction by aliphatic amine is not clearly described by FTIR spectra. No spectral assignments are reported, and the quality of the reported spectra is very poor. This part of the manuscript can be strongly improved.
4) the absence of some rheological data during the preparation of the mixture for nanocomposite preparation cannot be accepted. The rheological data are very important for a better understanding of the obtained results.
Author Response
Dear Editor
This is my response to your comments regarding our paper “Comparative Analysis of Properties of PVA Composites with Various Nanofillers: Pristine Clay, Organoclay, and Functionalized Graphene (nanomaterials-431778) in nanomaterials.
Thank you very much for the referee's comments. I have carefully revised the manuscript following the comments of the referee.
Response to Reviewer-3 comments
Point-1: The amount of 10% by wt is very huge for a nanocomposite.
Response-1: As the reviewer pointed out, the optimal amount could be 3% or 5% or less. Of course I agree completely with these points and I also have many of the same results. However, in the case of other polymers we have obtained, more is being reported. These results vary greatly depending on the type of filler added and how it is dispersed. For example, polyimide can be dispersed up to 40%. In this PVA study I wanted to see the results up to 10%.
Chang et al., Journal of Industrial and Engineering Chemistry, 2013, 19, 1593-1599.
Chang et al., Macromolecular Research, 2013, 21, 228-233.
Point-2: Explanation with the DSC thermograms.
Response-2: The DSC thermograms were supplemented to Fig-8 and the contents were explained. “The DSC thermograms of the PVA hybrids - with decreases in the peak intensity.”was added in the middle of Section E. Thermal properties.
Point-3: FT-IR
Response-3: The explanation of epoxy group was supplemented in detail, and the explanation of amine in HDA-GS was also explained in the picture and the text. “Figure 1 shows the FT-IR spectra - exhibits a peak at 1100 cm−1 (C-O stretching).” was added in Section A. FT-IR spectroscopy.
Point-4: Some rheological data during the preparation of the mixture for nanocomposite preparation
Response-4: Unfortunately, I could not test the rheological properties. Especially, in case of HDA-GS, since the amount of synthesis was too small (≤1 g), various experiments were not performed. I am so sorry for this.
I hope this revision is satisfactory for your further process.
Best,
Jin-Hae Chang
Professor

Reviewer 4 Report
The authors should try to theoretically describe the origin of the founded effects.
Author Response
Dear Editor
This is my response to your comments regarding our paper “Comparative Analysis of Properties of PVA Composites with Various Nanofillers: Pristine Clay, Organoclay, and Functionalized Graphene (nanomaterials-431778) in nanomaterials.
Thank you very much for the referee's comments. I have carefully revised the manuscript following the comments of the referee.
Response to Reviewer-4 comments
Point-1: Theoretical description be the origin of the founded effects.
Response-1: This paper has studied thermal properties and gas barrier. Therefore, the theory of gas barrier is described. “The mobility of the polymer chain - diffusion pathways through the nanocomposite [43,44].” was added in Section F.
I hope this revision is satisfactory for your further process.
Best,
Jin-Hae Chang
Professor

Round 2
Reviewer 2 Report
The article is ready for publication.
Reviewer 4 Report
I accept the article in the present form